# Inhibition of Liver Metastasis in Colorectal Cancer by Targeting IL-13/IL13Rα2 Binding Site with Specific Monoclonal Antibodies

**DOI:** 10.3390/cancers13071731

**Published:** 2021-04-06

**Authors:** Marta Jaén, Rubén A. Bartolomé, Carmen Aizpurua, Ángela Martin-Regalado, J. Ignacio Imbaud, J. Ignacio Casal

**Affiliations:** 1Department of Molecular Biomedicine, Centro de Investigaciones Biológicas, CSIC, Ramiro de Maeztu 9, 28039 Madrid, Spain; martajaen@cib.csic.es (M.J.); rubenabc@cib.csic.es (R.A.B.); angela.martin@cib.csic.es (Á.M.-R.); 2Protein Alternatives SL, Tres Cantos, 28760 Madrid, Spain; caizpurua@proteinalternatives.com (C.A.); jimbaud@proteinalternatives.com (J.I.I.)

**Keywords:** IL-13, IL13Rα2, SRC activation, therapeutic antibodies, metastasis, colorectal cancer

## Abstract

**Simple Summary:**

IL13Rα2 has been repeatedly reported as an excellent therapeutic target for multiple types of advanced cancers. However, previous IL13Rα2 targeting attempts have been mostly unsuccessful. Here, we describe a novel strategy based on the blocking of the IL-13 tumorigenic activity using a highly preserved D1 peptide selected from the IL13Rα2 binding site for mouse immunization and the inhibition of the cell invasion capacity for antibody screening. The IL13Rα2 D1 peptide-specific monoclonal antibody 5.5.4 has demonstrated a large capacity for blocking IL13Rα2 signaling capacity and protecting mice against established and non-established liver metastasis in colorectal cancer. These positive results predict a potential application to other IL13Rα2 positive cancers.

**Abstract:**

Background: IL13Rα2 is reportedly a promising therapeutic target in different cancers. Still, no specific antagonists have reached the clinics yet. We investigated the use of a IL-13/IL13Rα2 binding motif, called D1, as a new target for the development of therapeutic monoclonal antibodies (mAbs) for colorectal cancer (CRC) metastasis. Methods: IL13Rα2 D1 peptides were prepared and used for immunization and antibody development. Antibodies were tested for inhibition of cellular invasion through Matrigel using CRC cell lines. Effects of the mAbs on cell signaling, receptor internalization and degradation were determined by western blot and flow cytometry. Swiss nude mice were used for survival analysis after treatment with IL13Rα2-specific mAbs and metastasis development. Results: IL13Rα2 D1 peptides were used to generate highly selective mAbs that blocked IL13/IL13Rα2-mediated SRC activation and cell invasion in colorectal cancer cells. Antibodies also provoked a significant reduction in cell adhesion and proliferation of metastatic cancer cells. Treatment with mAbs impaired the FAK, SRC and PI3K/AKT pathway activation. Blocking effectivity was shown to correlate with the cellular IL13Rα2 expression level. Despite mAb 5.5.4 partially blocked IL-13 mediated receptor internalization from the cancer cell surface it still promotes receptor degradation. Compared with other IL13Rα2-specific antibodies, 5.5.4 exhibited a superior efficacy to inhibit metastatic growth in vivo, providing a complete mouse survival in different conditions, including established metastasis. Conclusions: Monoclonal antibody 5.5.4 showed a highly selective blocking capacity for the interaction between IL-13 and IL13Rα2 and caused a complete inhibition of liver metastasis in IL13Rα2-positive colorectal cancer cells. This capacity might be potentially applicable to other IL13Rα2-expressing tumors.

## 1. Introduction

Colorectal cancer (CRC) is among the four most lethal tumors worldwide [1]. Europe shows a high incidence, with the majority of its countries having a rate of 26 deaths per 100,000 people [1]. Mortality rate is very high among patients with metastasis and about 50% of the patients present with metastasis at diagnosis. Current treatment of metastatic CRC includes combinations of anti-EGFR monoclonal antibodies (mAbs) (cetuximab, panitumumab) or anti-VEGF (bevacizumab) with different FOLFOX and FOLFIRI regimes [2]. EGFR-based therapies suffer from low efficacy in those patients that present activating mutations in KRAS, BRAF and alterations in PIK3CA [3,4], and may develop some side effects at different extent. In patients with KRAS and BRAF mutations there is a limited clinical benefit from this therapy and the overall survival improvement is rather modest [2]. It has been estimated that near 80% of metastatic CRC patients do not benefit from anti-EGFR therapy [5]. Other promising immunotherapies as PD-1 blockade are also quite ineffective in CRC [6]. Therefore, patients will benefit from novel targets and novel strategies to develop therapeutic antibodies for a more successful treatment of disseminated colorectal cancer. 

IL-13 receptor α2 (IL13Rα2) is a cancer/testis-like tumor antigen [7], overexpressed in multiple tumors such as CRC, renal cell carcinoma, pancreatic, melanoma, head and neck, mesothelioma, ovarian cancer and glioblastoma, among others [8,9,10,11,12,13,14,15]. In CRC, higher expression was observed in T3 or T4 tumors as compared with T1 or T2 [16]. Initially considered a decoy receptor, we and others have demonstrated beyond reasonable doubt that IL13Rα2 is also a functional receptor for IL-13 signaling transmission in cancer cells [16,17,18,19]. The binding of IL-13 to IL13Rα2 triggers STAT6-independent cellular pathways, through different mediators including the phosphatase PTP1B and the scaffold protein FAM120A [19,20]. Indeed, the IL-13/IL13Rα2 signaling axis activates Src through PTP1B and then activates the Ras → Raf → MAPK cascade followed by the AP-1 transcriptional pathway in a number of human cancers [9,17,18,21]. As observed for many other receptors, IL-13 binding may induce receptor dimerization and/or internalization, where IL13Rα2 associates with multiple intracellular traffic proteins [20], being its recycling tightly controlled to regulate the surface expression levels and the amount of free receptor on the surface.

Due to its expression in advanced stages of cancer, IL13Rα2 has been postulated as a target for cancer therapy through multiple approaches (see [22] for a review). IL13Rα2 presents multiple advantages as therapeutic target; it is a signaling receptor specific of metastatic cancer cells, as it is only expressed in testis cells and some immune cells in adults [23]. IL13Rα2 does not present a significant mutation rate in colorectal cancer avoiding extensive testing on natural variants [24]. Blocking the IL-13/IL13Rα2 signaling capacity in metastatic cells should facilitate the development of new therapeutic strategies. Indeed, an IL13Rα2 D1 peptide located at the IL-13 binding site [25] showed strong inhibition capacity for the IL-13/IL13Rα2 signaling cascade in metastatic CRC and glioblastoma [26]. This 12 amino acid-long D1 peptide contains a lineal sequence (81-WKTIITKN-88) from IL13Rα2 that is highly preserved in many mammalian species. In this report, we hypothesized that D1 sequence-specific mAbs would combine all the therapeutic advantages of the mAbs in terms of specificity, selectivity and stability with the D1 blocking capacity. For comparative purpose, a D1-specific chicken antibody was prepared, as the low homology between both species would guarantee a stronger immunogenic response. Previous strategies to generate mAbs against IL13Rα2 have suffered from the high homology between human and mouse proteins. Basically, two alternatives have been explored. Either phage-display libraries were used for preparing antibody fragments that were conjugated to toxins [27] or IL13Rα2-specific mAbs were obtained against the recombinant protein [28]. These mAbs showed promising results in the survival of mice implanted with glioblastoma, although the mAb binding sequence was not identified [28].

Here, we have generated a panel of D1 sequence-specific monoclonal antibodies. One of them, mAb 5.5.4 exhibited a highly promising therapeutic effectivity as it was able to inhibit in vitro and in vivo the metastatic capacity of differentiated colorectal cancer cells. A potential application to other IL13Rα2-expressing tumors is foreseen.

## 2. Materials and Methods

### 2.1. Cell Lines, Reagents and Antibodies

The metastatic KM12SM colon cancer cell line was acquired from Dr. I. Fidler (MD Anderson Cancer Center, Houston, TX, USA). The cell lines SW480 and RKO were obtained from the ATCC and the SW620 cell line from the European Collection of Authenticated Cell Cultures (Salisbury, UK). HT-29 colon cancer cells were a kind gift from Dr. Mollinedo (CIB-CSIC). KM12SM were authenticated in our laboratory every 6 months. Murine CT26 colon carcinoma cell lines were obtained from Prof. Caroline Saucier (Université de Sherbrooke, Sherbrooke, QC, Canada). All cell lines were cultured as previously described [19]. Human IL-13 was used at 10 ng/mL and was purchased from PeproTech (London, UK). Irinotecan (Selleckchem, Munich, Germany) was used at 50 mg/Kg of weight. Antibodies used in the experiments are listed in Appendix A.

### 2.2. Peptide Design, Immunization, Preparation and Selection of Anti-IL13Rα2 Mouse Monoclonal Antibodies

Animal protocols for hybridoma generation were approved by the ethics committee of the Instituto de Salud Carlos III (CBA22_2014-v2) and Community of Madrid (PROEX 278/14). For mouse immunization, a 19-mer IL13Rα2 peptide (IGSETWKTIITKNLHYKD-Cys) was synthesized using F-moc Solid Phase Peptide Synthesis (Proteogenix, Schiltigheim, France) and conjugated to ovalbumin (OVA). The procedure to generate peptide-specific monoclonal antibodies was previously described [29]. Mouse serum and hybridoma clone selection were carried out by testing the antibody capacity to inhibit IL-13-triggered invasion in KM12SM cells. After clone selection, mAbs were purified using Protein G Sepharose antibody purification resin, and dialyzed against phosphate-buffered saline (PBS) for final testing and characterization. Isotype of final clones was determined with Rapid ELISA Mouse mAb Isotyping Kit (Pierce, Madrid, Spain).

### 2.3. Chicken Antibodies

Alternatively, a 13-mer IL13Rα2 synthetic peptide (GSETWKTIITKNC) (Proteogenix) was further conjugated to OVA. All animal experiments in this study were conducted according to the European Union Directive 2010/63/EU. Two (2) White Leghorn chickens were immunized and boosted under the wing, in the armpit, during 9 weeks with the OVA-conjugated IL13Rα2 peptide according to standard procedures. Purification of chicken polyclonal antibodies from collected eggs was performed by precipitation with sodium sulfate followed by affinity chromatography against the IL13Rα2 13-mer peptide of the IgY fraction, to finally obtain the specific chicken anti-IL13Rα2 antibodies. Purified antibodies were dialyzed against PBS for final testing and usage.

### 2.4. Capture ELISA

Maxisorp 96-well microtiter plates (Nunc, Roskilde, Denmark) were coated by addition of 100 μL/well of a 5 μg/mL solution of anti-IL13Ra2 mAbs 4.4.2 or 5.5.4 in carbonate buffer (50 mM, pH 9.6) and overnight incubation at 4 °C. Coated plates were washed three times with washing buffer (PBS with 0.05% (*v*/*v*) Tween-20) and then 150 μL of 2% BSA in PBS was added to each well to reduce non-specific binding. The plate was washed three times with washing buffer and 100 μL of Biotin-GGGSETWKTIITKN peptide was added to coated wells in log_2_ dilution from a 2 μg/mL standard. An irrelevant biotin labelled-protein was used as control. Plates were incubated at 37 °C for 1 h and washed three times with washing buffer. Next, 100 μL of a 1/2000 dilution of streptavidin-HRP conjugated (Southern Biotechnology, Birmingham, AL, USA) was added to each well and plates were incubated for 30 min at RT. Finally, plates were washed 5 times with washing buffer and 100 μL/well of TMB (3,3′,5,5′-tetramethylbenzidine substrate) substrate solution (Sigma Aldrich, Tres Cantos, Spain) was added. Color development was stopped after 10 min incubation by adding 50 μL/well of H_2_SO_4_ (2N). The absorbance was measured at 450 nm. To further assess the specificity against IL13Rα2, microtiter plates (Maxisorp, Nunc) were coated with 1 μg/mL of IL13Rα2-derived D1-short (D1S) peptide (WKTIITKN) or IL13Rα1-binding site derived Rα1 peptide (KQDKKIAPE). After washing with PBS, plates were blocked with 3% skimmed milk in PBS for 2 h at room temperature. Then, mAbs at different concentrations (0–5 µg/well) were added to the plates and incubated for 1.5 h at room temperature. An irrelevant antibody was used as a control. HRP-anti-mouse IgG (Thermo Scientific, Madrid, Spain) was added for 1.5 h at room temperature. The absorption was measured as before.

### 2.5. Competition ELISAs

Competition ELISA was carried out as previously described [26]. Briefly, plates (Maxisorp, Nunc) were coated overnight with 1 μg/mL of IL13Rα2 (Protein Alternatives SL, Tres Cantos, Spain). IL-13 (0.1 μg/mL) and mAbs (including a control antibody) at different concentrations (1–50 µg/well) were added to the plates and incubated for 1.5 h at room temperature. After washing, peroxidase-labeled anti-IL-13 antibody (Abexxa, Arlington, TX, USA) (0.4 μg/mL) was added for 1.5 h at room temperature. Then, the signal was developed with TMB (Sigma-Aldrich) and the reaction stopped with 1 M HCl. Absorption was measured at 450 nm.

To test the D1 peptide specificity of the antibodies, microtiter plates were coated, washed and blocked as before. Then, mAbs (50 μg/mL) together with D1S (WKTIITKN) and Ala-variants (D1-4A (WKTAITKN) and D1-5A (WKTIATKN)) peptides [26] at different concentrations (1–10 µg/mL) were added to the plates and incubated for 1.5 h at room temperature. An irrelevant antibody was used as a control. After washing, HRP-anti-mouse IgG (Thermo Scientific) was added for 1.5 h at room temperature and the absorption measured as before.

### 2.6. Antibody Confocal Microscopy

Human colorectal KM12SM, KM12C, SW620 and SW480 cells were seeded onto Matrigel-coated cover slides. After reaching near confluence, cells were fixed with 4% paraformaldehyde in PBS. After three washes, cells were incubated with primary and control antibody (30 µg/mL) in PBS containing human gamma-globulin (40 µg/mL) at 4 °C overnight. Then, cells were made to react with Alexa-488-labelled secondary antibodies and 4,6-diamidino-2-phenylindole (DAPI) for 40 min at room temperature. Finally, slides were treated with Mounting Fluorescence Medium (Dako, Glostrup, Denmark) and images were obtained with a TCS-SP5-AOBS confocal microscope (Leica Microsystems, Wetzlar, Germany).

### 2.7. Flow Cytometry

Flow cytometry analysis was performed as previously described [26].

### 2.8. Western Blot Analysis and Immunoprecipitation 

For western blotting, cells were removed, washed with cold PBS and lysed with lysis buffer (1% Igepal, 50 mM NaCl, 2 mM MgCl_2_, 10% glycerol, containing protease (Roche, Basel, Switzerland) and phosphatase inhibitors (Sigma-Aldrich) in water. Protein extracts were run in 10% SDS-PAGE and transferred to nitrocellulose membranes following standard procedures. Membranes were incubated with primary antibodies (Appendix A) and then with either HRP-anti-mouse IgG (Thermo Scientific) or HRP-anti-rabbit IgG or HRP-anti-goat IgG (Sigma-Aldrich). Immunodetection was achieved with SuperSignal West Pico Chemiluminescent Substrate (Thermo Scientific). For immunoprecipitation, 500 μg of cell lysates were incubated with 25 μL of Protein G-sepharose beads (Sigma-Aldrich) for 4 h at 4 °C to remove unspecific binding to beads. Then, supernatants were incubated for 16 h at 4 °C with 3 µg of the indicated antibodies in presence of 25 μL of Protein G-sepharose beads. After four washes (100 g, 1 min), the immunocomplexes were resuspended in Laemmli buffer, boiled for 5 min and loaded onto SDS-PAGE for western blot analysis. As a control, we incubated the lysates with an irrelevant antibody to discard unspecific binding to immunoglobulins. The original western blotting figures can be found in Appendix A.

### 2.9. IL13Rα2 Internalization and Degradation

KM12SM cells were kept in serum-free DMEM for 3 h, detached using 2mM EDTA in PBS and incubated at 37 °C for 45 min in the presence of IL-13 (10 ng/mL) with or without the indicated antibodies (15 μg/mL). After washing, cells were incubated at 4 °C for 1 h with anti-IL13Rα2 antibodies, followed by HRP-conjugated anti-mouse Ig for 30 min at 4 °C. Fluorescence intensity was measured in a cell fluorimeter as previously described [26]. Receptor degradation was analyzed by western blot using duplicate analysis.

### 2.10. Cell Signaling

Cells were incubated for 3 h in serum-free DMEM, detached with 2mM EDTA, washed and treated with anti-IL13Rα2 D1 mAbs (15 μg/mL) for 15 min at 37 °C and then treated with IL-13 (10 ng/mL) at different times (0, 10, 60 min). Ice cold PBS was used to stop the reaction and, finally, cells were lysed and subjected to western blot as before

### 2.11. Cell Adhesion and Proliferation

Cancer cells were kept in serum-free medium for 3 h, detached with 2 mM EDTA in PBS and labelled with BCECF-AM (Molecular Probes, Eugene, OR, USA). Following a 10-min incubation in serum-free medium with anti-IL13Rα2 peptide mAbs (15 μg/mL) and/or IL-13 (10 ng/mL), 6 × 10^4^ cells were loaded into 96-well plates previously coated with Matrigel (0.4 μL/well) (BD Biosciences, Madrid, Spain) and blocked with 0.5% BSA. After 25-min incubation of the loaded plates at 37 °C, non-adhesive cells were removed by three washes with serum-free medium. Adhesive cells were lysed with 1% SDS in PBS and cell adhesion was quantified in a POLARstar Galaxy fluorescence analyser (BMG Labtech, Ortenberg, Germany). Cell proliferation assays were carried out as previously described [26].

### 2.12. Wound Healing

1 mm-wide scratch was made across a confluent monolayer of cancer cells in 24-well plates previously coated with Matrigel (0.2 μg/well). Plates were incubated in 0.5% serum medium with anti-IL13Rα2 peptide mAbs (15 μg/mL) and/or IL-13 (10 ng/mL) for 24 h at 37 °C. Distance covered by cells was estimated using pictures of the scratches taken at times 0 and 24 h. Migration speed was obtained as the distance covered divided by 24 h and divided by 2 (as cell migrated by two sides of the wound).

### 2.13. Cell Invasion Assays

For invasion assays, 6.5 mm diameter Transwell filters with 8 μm pores (Corning Inc., Glendale, AZ, USA) were filled with a mix of 22 µL of serum-free DMEM and 11 µL of Matrigel (BD Biosciences). When Matrigel solidified, 6 × 10^4^ cells in 200 µL serum-free DMEM were loaded in the upper compartments of Transwells in the presence of anti-IL13Rα2 peptide or control mAbs (15 μg/mL); whereas the lower compartments were filled with IL-13 (10 ng/mL) in 700 µL of serum-free DMEM. After 48 h, cells which did not reach the lower surface of the filters were removed, while migrated cells were fixed with 4% paraformaldehyde, dyed with crystal violet, and observed under a microscope. The number of invasive cells to IL-13 in presence of control antibodies was considered as 100% of IL-13-triggered cell invasion.

### 2.14. Experimental Metastasis in Nude Mice

Animal experiments were approved by the Ethics Committees of the CSIC and Community of Madrid (PROEX 252/15) and performed as previously described [29]. Briefly, Swiss nude mice (*n* = 6 per condition) were inoculated intra-splenically with 1.5 × 10^6^ KM12SM cells in 0.1 mL PBS. Twenty four hours later spleens were removed. Then, these mice were treated intravenously with anti-IL13Rα2 mAbs 5.5.4, 4.4.2 or vehicle control. Antibodies were used at 50 mg/Kg of weight in a total of 7 doses, starting 2 or 7 days after inoculation for the immediate or established metastasis models, respectively. Mice survival rates were recorded twice a week. Then mice were subjected to necropsy and inspected for metastasis in liver. Liver homing experiments were performed as previously described [26]. For lung metastasis, mice were inoculated in the tail vein with 1 × 10^5^ mouse CT-26 cells in 0.1 mL PBS and treated with the 5.5.4 mAb or vehicle control as before. Irinotecan treatment (50 mg/Kg of weight) was divided in 4 doses during 2 weeks). Mice survival rates were followed as before.

### 2.15. Statistical Analyses

Data from in vitro assays were analyzed by one-way ANOVA followed by Tukey-Kramer multiple comparison tests and the average represented in histograms, showing the standard deviation as error bars. For in vivo assays, Kaplan-Meier survival curves were analyzed by log-rank test. In all analyses, a value of *p* < 0.05 was considered statistically significant.

## 3. Results

### 3.1. Antibody Selection with the Capacity for Blocking IL13Rα2-Mediated Invasion

Mice were immunized with a 19-mer D1-derived IL13Rα2-peptide coupled to OVA (Appendix A). For hybridoma selection, a functional screening was implemented to test the antibody capacity for blocking the IL-13/IL13Rα2-promoted cell invasion through Matrigel. As a first step, pre-hybridoma fusion mouse sera were tested. Serum #2 showed a 40% inhibition of the cellular invasive capacity (Figure 1A). This mouse was selected for spleen fusion and antibody production after a final boost with a recombinant IL13Rα2-Fc protein (Appendix A). Then, 20 antibody clones were screened by their capacity to inhibit invasion (Figure 1B) and peptide recognition using ELISA (Figure 1C). Two hybridoma supernatants (#4–5) inhibited >80% invasion of KM12SM cells and were selected for further subcloning. ELISA testing against the uncoupled peptide offered significant binding values for both supernatants. After further subcloning, clones 4.4 and 5.5 inhibited invasion by more than 50% and were selected for final clone selection (Figure 1D). Final testing of eight clone supernatants resulted in the selection of two clones, 4.4.2 and 5.5.4, that inhibited IL-13-mediated cell invasion more than 60% in KM12SM and SW620 (Figure 1E). These two mAbs were isotyped (4.4.2: IgG2b/κ, 5.5.4: IgG1/κ) and used for further experiments. Chicken antibody (GC-13) prepared against the same peptide also exhibited strong invasion inhibition (Figure 1F).

### 3.2. Characterization of IL13Rα2 D1-Specific mAbs

MAbs 4.4.2 and 5.5.4 specificity was confirmed by using different assays. A capture ELISA using the mAbs coated at 5 µg/mL selectively bound the biotinylated D1 peptide vs an irrelevant peptide. A significant capture up to 250 ng/mL was observed for the biotinylated peptide respect to the control peptide for both mAbs (Figure 2A). Furthermore, we performed a competition assay between the two mAbs and the recombinant IL-13 for the binding to the IL13Rα2 receptor. MAbs blocked IL-13 binding to IL13Rα2 starting at 5 µg/well, with a complete inhibition at 50 µg/well (Figure 2B). A control antibody only exhibited a minor competition (<20%) at high concentrations. Next, we tested the specificity of the mAbs towards the IL13Rα2 D1 peptide: first, we verified that only IL13Rα2-derived D1 peptides can displace the mAb from binding to IL13Rα2 ectodomain (Figure 2C). As expected, only peptide D1S, as well as alanine variants (D1-4A, D1-5A) were able to displace the antibody binding, suggesting that 5.5.4 and 4.4.2 tolerate punctual variations in the recognized epitope. Then, we tested the binding to an IL13Rα2-derived peptide (D1S) compared to the binding to an IL13Rα1-derived peptide (Rα1), within the motif sequences bound by IL-13 in this receptor). MAbs exclusively bound to IL13Rα2 D1 peptide (Figure 2D). Both mAbs were negative by western blot, but positive for IL13Rα2 immunoprecipitation (Figure 2E). Flow cytometry results were negative except for a weak reactivity of the 4.4.2 mAb with KM12SM cells when compared to the chicken antibody and a commercial antibody (Figure 2F). Finally, we tested the reactivity of mAbs 4.4.1, 4.4.2, 5.5.4 and chicken antibodies with different CRC cell lines using confocal microscopy. Commercial antibody 2K8 was used as positive control. Whereas mAbs 4.4.1 and 4.4.2 and commercial antibody exhibited membrane and cytoplasmic staining in all cell lines (Figure 2G), 5.5.4 was negative. In contrast, chicken antibody was more membrane-specific than 4.4.2. In summary, IL13Rα2 D1 peptide-specific mAbs exhibited a relatively weak reactivity with IL13Rα2 using conventional techniques. However, they appear to recognize an IL13Rα2 conformation required for cell invasion, which would confer a high functional selectivity to their use. Therefore, mAbs 4.4.2 and 5.5.4 were selected for further testing of their neutralizing capacity.

### 3.3. IL13Rα2 D1-Specific mAbs Inhibit Metastatic Properties in Colorectal Cancer Cells

Next, we examined the capacity of mAbs 4.4.2 and 5.5.4 to inhibit IL13-mediated metastatic capacity in KM12SM and SW620 cells, which differ in the levels of cell membrane IL13Rα2 expression (KM12SM < SW620, Figure 2F) and phenotypic properties (epithelial vs. mesenchymal). Overall, there was a significant similarity between the results obtained in both cell lines (Figure 3). Both mAbs inhibited at a similar extent cell adhesion, migration, invasion and proliferation in the two metastatic colorectal cancer cell lines. Interestingly, GC-13 chicken antibody prepared against the same IL13Rα2 peptide also showed excellent blocking properties for IL-13 mediated metastatic capacity (Figure 3). Together, these results support the capacity of D1-specific mAbs for blocking IL-13-triggered cell invasion, migration and adhesion in colorectal cancer cell lines positive for IL13Rα2 expression.

### 3.4. IL13Rα2 D1-Specific mAbs Inhibit Ligand-Induced Phosphorylation of Downstream Signaling Molecules

To further explore the alterations in the IL-13/IL13Rα2 signaling pathway, human KM12SM, SW620, RKO and murine CT-26 cell lines were treated with the two mAbs or a control IgG. These cell lines were chosen because they exhibited different levels of IL13Rα2 expression. RKO and mouse CT-26 cell lines are aggressive, poorly differentiated cell lines characterized by the lack of IL13Rα2 membrane expression [16,30]. In KM12SM cells, a swift activation of pFAK and pSrc (10 min) was observed, followed by slower pAKT and pERK activation (60 min).

Both mAbs diminished the activation of FAK, Src, AKT and ERK kinases (Figure 4), which correlate with the observed decrease in cell adhesion, invasion and proliferation. In SW620 cells, FAK, Src and AKT inhibition occurred after a late activation at 60 min. No activation of pERK was observed in SW620 and RKO cells, which might be explained by the presence of KRAS and BRAF mutations in SW620 in RKO, respectively [31]. As expected, due to the lack of IL13Rα2 expression, RKO and murine CT-26 cells did not respond to IL-13 in the 60 min period. Therefore, no effect was observed for the mAbs on IL-13-mediated signaling in RKO and CT-26 cells. In summary, mAbs inhibited FAK, Src and AKT signaling after blocking the IL-13/IL13Rα2 axis in the metastatic cell lines expressing IL13Rα2. Downstream signaling inhibition correlates with the initial levels of IL13Rα2 membrane expression and the presence of mutations in the RAS-RAF-ERK cascade.

### 3.5. IL13Rα2 D1 mAbs Inhibit IL13Rα2 Activation But Not IL13Rα1 Signaling

IL-13 signaling pathway involves the use of two different receptors. The canonical IL-13 receptor involves a heterodimer of IL13Rα1 with IL4Rα and signaling occurs through STAT6 activation, whereas IL13Rα2-mediated signaling is SRC-mediated, STAT6-independent. To examine the selectivity of the antibody interaction, we investigated the specificity of 4.4.2 and 5.5.4 to block both IL-13 receptors through the inhibition of the SRC and STAT6 activation pathways in the HT-29 cell line, which expresses both receptors in contrast to KM12SM or SW620 cell lines [26].

Both antibodies inhibited SRC, but not STAT6 activation (Figure 5A). Therefore, mAbs 4.4.2 and 5.5.4 selectively block the IL-13 binding to IL13Rα2 but not to IL13Rα1. This specificity might be highly relevant for those diseases that involve IL13Rα2 signaling but not IL13Rα1.

### 3.6. IL13Rα2 D1 mAbs Prevent the Internalization Capacity of IL-13 Binding to IL13Rα2

Receptor internalization is involved in signaling regulation. To test this notion, we explored the effect of mAbs on the IL13Rα2 internalization and degradation promoted by IL-13 using flow cytometry and western blot. IL-13 triggered a significant internalization of IL13Rα2 that was partially inhibited after treatment with D1-specific antibodies (Figure 5B). However, both mAbs induced certain degree of receptor internalization. Regarding receptor degradation, we found some differences after antibody treatment. So, 5.5.4-treated cells showed more receptor degradation than 4.4.2-treated cells in KM12SM cells (Figure 5C). This effect was more visible in KM12SM cells than in SW620 probably due to the higher levels of IL13Rα2 membrane expression in SW620 and the presence of different adaptors. Therefore, mAb 5.5.4 seems to play a dual effect on IL13Rα2. On the one hand, 5.5.4 exhibits an antagonist activity on IL-13 binding, but on the other appears to cause an agonist-like effect after antibody binding, which induces a direct degradation of the receptor without activation.

### 3.7. Monoclonal Antibody 5.5.4 Is a Potent Inhibitor of Colorectal Cancer Metastasis

Finally, we evaluated the capacity of mAbs 4.4.2 and 5.5.4 to protect mouse against colorectal cancer metastasis. Swiss nude mice were inoculated into the spleen with KM12SM cells to induce liver colonization through the hepatic portal vein. Then, spleens were removed and mice (*n* = 6) were treated with each mAb individually for 2 weeks, with a total dose of 50 mg of antibody per kg of mouse weight. Kaplan-Meier survival results indicated a quite different response to the antibody treatment, despite both mAbs were equally effective on blocking invasion capacity. Whereas mAb 4.4.2 slightly improved mice survival respect to the control group, mAb 5.5.4 conferred a strong protection, with 80% of the mice surviving the end-point without apparent metastatic lesions (Figure 6A). To further confirm the neutralizing activity of mAb 5.5.4, we performed a second in vivo experiment. All mice treated with the mAb 5.5.4 survived to the inoculation of metastatic KM12SM cells without development of liver metastatic nodules (Figure 6B). Then, we investigated the underlying mechanism of 5.5.4 to inhibit liver metastasis. Cancer cells were inoculated in the spleen in the presence of 5.5.4 or a control antibody. After 4 days, the presence of cancer cells in the liver was determined by PCR of a surrogate human gene (GAPDH). Livers obtained from mice inoculated with the 5.5.4 showed negligible amounts of human GAPDH compared to those inoculated with the control antibody (Figure 6C), indicating that 5.5.4 mAb blocks the cell homing and the metastatic colonization of colon cancer cells. Furthermore, to demonstrate that this protection was related to the IL13Rα2 expression levels, we tested intravenous inoculation of CT-26 for lung metastasis treatment. Since these cells do not express IL13Rα2, after using either mAb 5.5.4 or irinotecan as treatments, mice survival to lung metastasis was not significantly improved (Figure 6D). To note that irinotecan, a drug widely used in CRC chemotherapy, also failed to provide significant protection against CT-26 cells. Finally, we tested the effectiveness of 5.5.4 in established liver metastasis. In this experiment, antibody treatment started 7 days after the intra-splenic inoculation of the cells. All mice receiving the mAb 5.5.4 survived without metastasis development (Figure 6E). In summary, mAb anti-IL13Rα2 5.5.4 has shown the capacity to protect mice against IL13Rα2 positive differentiated liver metastasis, including established metastasis. These results suggest a potential therapeutic application of mAb 5.5.4 in colorectal cancer metastasis.

## 4. Discussion

There is a strong necessity to reduce metastasis-associated mortality in colorectal cancer. Multiple studies have demonstrated the value of IL13Rα2 as a promising target for different types of metastasis, including colorectal cancer. In this report, we have developed a panel of mAbs specific for the IL-13 binding site in IL13Rα2, using the D1 sequence as antigen [26]. Antibodies were selected according to their capacity for invasion inhibition. ELISA experiments confirmed that mAbs 4.4.2 and 5.5.4 blocked the binding of IL-13 to IL13Rα2, whereas D1 peptides inhibited the antibody binding to IL13Rα2. Moreover, 4.4.2 and 5.5.4 antibodies were highly specific of IL13Rα2 as they did not recognize IL13Rα1 peptides from the binding site nor reacted with the complete IL13Rα1. In functional analyses, both mAbs inhibited IL-13 pro-tumorigenic properties, including migration and invasion. Antibody effects were mediated through the inhibition of FAK, SRC, AKT and other IL13Rα2 downstream signaling mediators’ activation. However, mAbs did not affect STAT6 signaling, IL13Rα1-mediated, confirming that blocking activity was IL13Rα2-specific. Mab 5.5.4 also played a dual effect on receptor internalization and degradation as antagonist and agonist. Finally, we also provide direct evidence that treatment with mAb 5.5.4 caused a complete inhibition of liver metastasis in established and non-established liver metastasis. This inhibition is likely caused by blocking cell homing in the liver and further cell expansion. No side effects on weight or overall health status were observed in the treated mice. Taken together, these observations indicate a strong potential therapeutic value for mAb 5.5.4.

The anti-metastatic efficacy of mAb 5.5.4 was probably achieved as a consequence of the functional screening of the hybridoma supernatants for invasion inhibition. In contrast to a conventional ELISA screening, our strategy facilitates the selection of mAbs with functional blocking properties [29]. However, this strategy also presents some limitations. Indeed, 5.5.4 and 4.4.2 displayed a highly specific but weak reactivity with IL13Rα2 using conventional techniques. The narrow reactivity of these mAbs may indicate the recognition of a transient native conformation (i.e., that used during the receptor interaction). In any case, the high selectivity for activated-only IL13Rα2 might turn out to be beneficial, avoiding indiscriminate targeting and minimizing negative collateral effects. Indeed, high affinities can be suboptimal for therapeutic antibodies that target solid tumors, as affinity increases, tumor penetration of the antibody decreases as well as tumor uptake [32,33]. Lastly, if necessary, affinity might be improved using different in vitro strategies [34].

Whereas KM12SM cells give place to well-differentiated, colon-like tumors [31], SW620 and RKO have been described as undifferentiated, mesenchymal cell lines [31], the same as poorly differentiated mouse CT-26 cells [30]. The low expression level of IL13Rα2 in RKO [16] and CT-26 is rather surprising as IL13Rα2 overexpression is associated with more aggressive, mesenchymal cancer phenotypes and we have previously found a significant association between high IL13Rα2 expression and high grade and colloid tumors [16]. Indeed, IL13Rα2 is considered a cancer/testis antigen, which are usually associated to tumors of high histological grade and late clinical stages [35]. Interestingly, our results demonstrate the capacity of 5.5.4 to neutralize IL13Rα2-positive metastasis coming from differentiated cell lines.

Both mAbs inhibited FAK, SRC and AKT activation in KM12SM and SW620 cell lines. However, we observed some differences in signaling kinetics after ligand activation according to the IL13Rα2 expression level. Higher expression correlates with a faster activation. We also noticed that both mAbs were able to inhibit cell proliferation independently from the presence of KRAS mutations, as occurs in SW620 cells. This result is stimulating as many colorectal cancers contain KRAS mutations, which render ineffective other targeted therapies such as those based on EGFR-treatments. Mabs were specific for blocking only IL13Rα2 activation in positive tumors, excluding collateral effects on IL13Rα1-mediated STAT6 activation. Interestingly, this high selectivity might avoid interferences with IL-4/IL-13 physiological effects through IL-4Rα/IL13Rα1. These results also confirm that the pro-metastatic activity of IL-13 is completely IL13Rα2-dependent, as IL13Rα1-mediated signaling was still active after metastasis inhibition.

Minor differences in epitope recognition or receptor internalization and degradation might have a major effect in the different capacity of 4.4.2 and 5.5.4 for blocking liver metastasis; despite they are functionally equivalent in many other aspects, including invasion inhibition. Mab 5.5.4 appears to be more effective than 4.4.2 in blocking IL-13-promoted IL13Rα2 internalization and recycling. Therefore, 5.5.4 seems to present a combination of direct antagonistic effect, blocking the receptor activation, with a simultaneous antibody-promoted receptor internalization and degradation (agonist effect). Endocytic trafficking appears to enable specific signaling pathways from intracellular sites and might play a critical role not only in attenuating IL13Rα2 signaling but also in controlling specific pathways [36]. The antibody isotype is another important difference between 5.5.4 (IgG1) and 4.4.2 (IgG2b), as it plays a key role in some immune effector functions (i.e., antibody-dependent cellular cytotoxicity (ADCC)) [37,38]. IgG1 molecules have shown more ADCC capacity that IgG2 antibody. This might suggest a potential role for ADCC in the 5.5.4 effects, as ADCC has demonstrated to have a major role in antibody therapeutic efficacy [38]. This potential capacity deserves further investigation. Altogether, these properties might explain the superior anti-metastatic capacity of mAb 5.5.4.

We could not test the effect of 5.5.4 on SW620 cells in vivo. These cells were derived from a lymph node metastasis but they are quite inefficient in causing liver metastasis. The poor effectivity of 5.5.4 in preventing lung metastasis caused by mouse CT-26 cells is likely explained by the lack of IL13Rα2 expression in these cells [30] and the different regulation of IL-13 bioactivity between rodent and human cells [39]. Mouse cells express a soluble form of IL13Rα2, which might interfere with the antibody to deplete extracellular IL-13, as secreted IL13Rα2 may bind the antibody in the circulation and could prevent sufficient antibody from binding to the tumor. Based on its molecular signature, CT-26 cell line was predicted to be also refractory to anti-EGFR mAbs [30]. Irinotecan, a widely used compound in the treatment of advanced CRC, was also ineffective against CT-26 metastasis. The low efficacy of both compounds for the treatment of CT-26 metastasis discouraged us from trying a combination therapy in this case.

Currently used targeted EGFR therapy is only effective in patients with wild type RAS. IL13Rα2 was described to be upregulated following activation of EGFR and mutant EGFRvIII in cancer cells [40]. However, we have recently demonstrated that IL-13 signaling is independent of EGF signaling, although both pathways require PTP1B for Src activation [19]. In the case of IL13Rα2, Src activation promotes proliferation, migration and invasion [19,20], whereas EGFR activation mainly promotes cancer cell proliferation. Based on these findings, we can think that IL13Rα2-specific mAbs might be used in combination with or alternatively to those mAbs currently used for EGFR targeting in colorectal cancer [41]. As tumors develop resistance to EGFR-based therapy though different mechanisms [38], IL13Rα2 therapy might be an alternative to overcome EGFR-based resistance.

## 5. Conclusions

In summary, we have obtained an IL13Rα2-specific antibody 5.5.4 that can inhibit IL-13-mediated FAK, Src and AKT signaling to suppress metastatic liver colonization in colorectal cancer. The complete mouse survival indicates a potential therapeutic effect in colorectal cancer metastasis, including established and differentiated metastasis. Given the significant expression of IL13Rα2 in other late-stage human tumors (ovarian, pancreatic, glioblastoma), we believe that 5.5.4 mAbs might be clinically useful for these other human tumors.

## Figures and Tables

**Figure 1 cancers-13-01731-f001:**
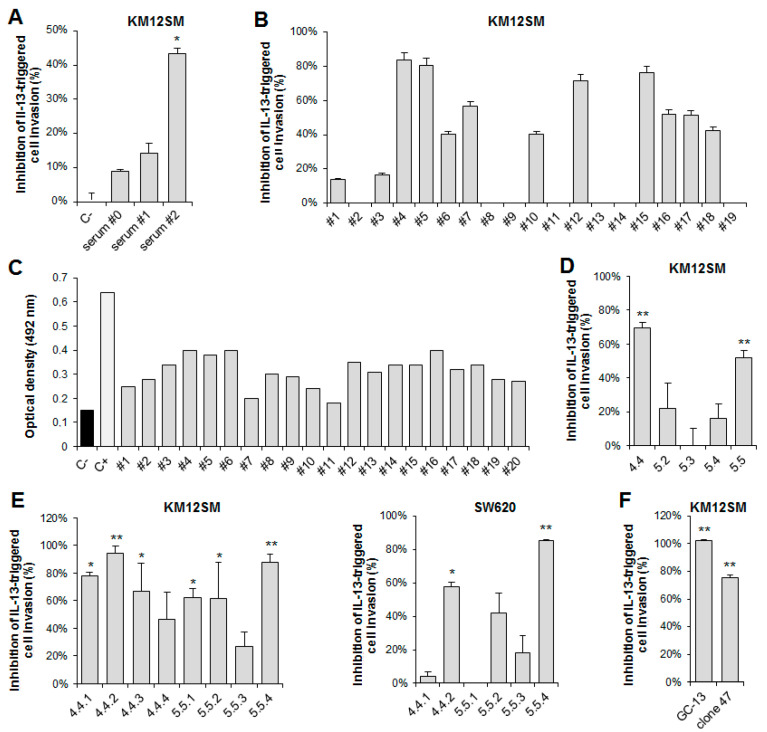
Functional screening and selection of IL13Rα2 D1-specific antibodies to inhibit IL13-promoted cell invasion. For antibody screening, KM12SM cells were subjected to inhibition of cell invasion through Matrigel in the presence of (**A**) the polyclonal serum of each animal and (**B**) supernantants after hybridoma fusion. (**C**) Same supernatants were tested by indirect ELISA against the uncoupled IL13Rα2 peptide. (**D**,**E**) Preselected clones #4 and #5 and selected subclones were tested in cell invasion inhibition assays either in KM12SM or SW620 cells. (**F**) Cell invasion assays with a chicken antibody (GC-13) prepared against a similar D1 peptide. * *p* < 0.05; ** *p* < 0.01.

**Figure 2 cancers-13-01731-f002:**
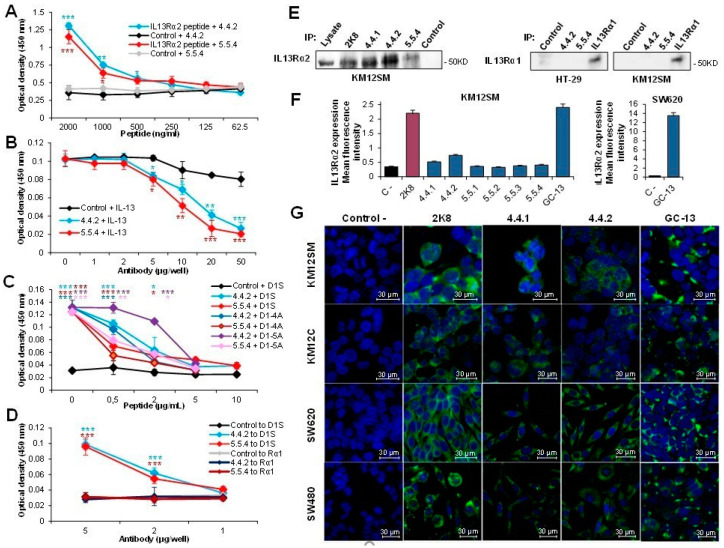
Functional characterization of anti-IL13Rα2 D1 peptide antibodies. (**A**) Capture ELISA of IL13Rα2 D1 biotin-labelled peptide or an irrelevant biotin-labelled peptide using immobilized anti-IL13Rα2 mAbs 4.4.2 or 5.5.4. Quantification of bound peptide was performed by streptavidin-HRP. MAbs 4.4.2 and 5.5.4 significantly bound to IL13Rα2-derived peptide compared to an irrelevant peptide (* *p* < 0.05; ** *p* < 0.01; *** *p* < 0.001). (**B**) MAbs 4.4.2 or 5.5.4 significantly inhibited the binding of IL-13 to immobilized IL13Rα2 using a competition ELISA (* *p* < 0.05; ** *p* < 0.01; *** *p* < 0.001). (**C**) IL13Rα2-derived peptides (D1S and alanine variant forms of D1) inhibited the binding of mAbs 4.4.2 or 5.5.4 to immobilized IL13Rα2 using a competition ELISA. MAbs 4.4.2 or 5.5.4 significantly bound to IL13Rα2 compared to an irrelevant antibody (* *p* < 0.05; ** *p* < 0.01; *** *p* < 0.001). (**D**) MAbs 4.4.2 or 5.5.4 significantly bound to IL13Rα2-derived peptide (D1S) compared to IL13Rα1-derived peptide (* *p* < 0.05; ** *p* < 0.01; *** *p* < 0.001). (**E**) IL13Rα2 immunoprecipitation using IL13Rα2 D1 mAbs in KM12SM cells (2K8, commercial IL13Rα2 Ab, Ctrl: Negative control). As positive control a lane with 100 μg of whole cell lysate of KM12SM cell was added to the blot. The indicated antibodies were also tested for immunoprecipitation of IL13Rα1 in HT-29 and KM12SM extracts. Only commercial anti-IL13Rα1 was able to immunoprecipitate this receptor. (**F**) Flow cytometry using IL13Rα2 D1 mAbs. (**G**) Immunofluorescence analysis of IL13Rα2 expression and staining using IL13Rα2 D1-specific mAbs in KM12SM, KM12C, SW620 and SW480 colorectal cancer cells. Scale bar: 30 µm.

**Figure 3 cancers-13-01731-f003:**
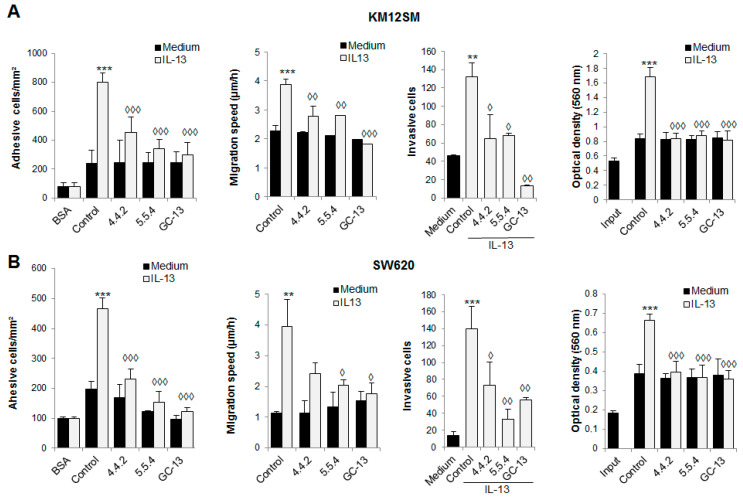
Inhibition of IL-13 prometastatic capacities. (**A**,**B**) Metastatic KM12SM and SW620 colorectal cancer cells were treated with IL-13 (10 ng/mL) and subjected to cell adhesion, wound healing, invasion and MTT assays in the presence of the anti-IL13Rα2 D1-specific mAbs, chicken antibody GC-13 or a control antibody. The presence of IL-13 significantly increase cell invasion, adhesion, migration and proliferation (** *p* < 0.01; *** *p* < 0.001), whereas anti-IL13Rα2 D1 peptide antibodies significantly inhibited these pro-metastatic capacities triggered by IL-13 (◊ *p* < 0.05; ◊◊ *p* < 0.01; ◊◊◊ *p* < 0.001). In proliferation assays, “input” means cells collected “ab initium”, i.e., cells allowed to adhere to the plate for 1 h at 37 °C and then incubated for 1 h with MTT before absorbance was determined. For the other conditions, cells were incubated with 1% serum for 48 h at 37 °C and then incubated with the MTT reagent.

**Figure 4 cancers-13-01731-f004:**
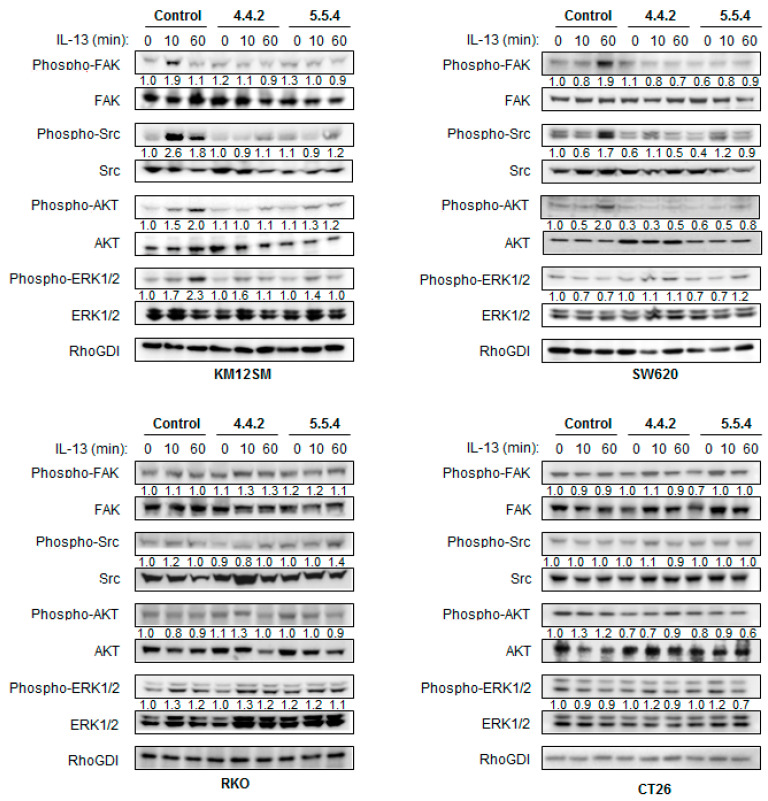
Molecular pathway analysis of cell signaling inhibition by IL13Ra2 D1-specific antibodies. KM12SM, SW620, RKO and CT-26 colorectal cancer cell lines were treated with IL-13 for the indicated times in serum-free DMEM in absence or presence of IL13Rα2 D1-specific mAbs. Cell extracts were collected at the indicated times and analyzed by western blot with antibodies against FAK, SRC, AKT, ERK1/2 and their phosphorylated forms. RhoGDI was used as a loading control. Bands were quantified and normalized with respect to untreated condition. IL13Rα2 D1-specific antibodies inhibited phosphorylation of IL-13/IL13Rα2 signaling mediators in a time and cell type-dependent mode.

**Figure 5 cancers-13-01731-f005:**
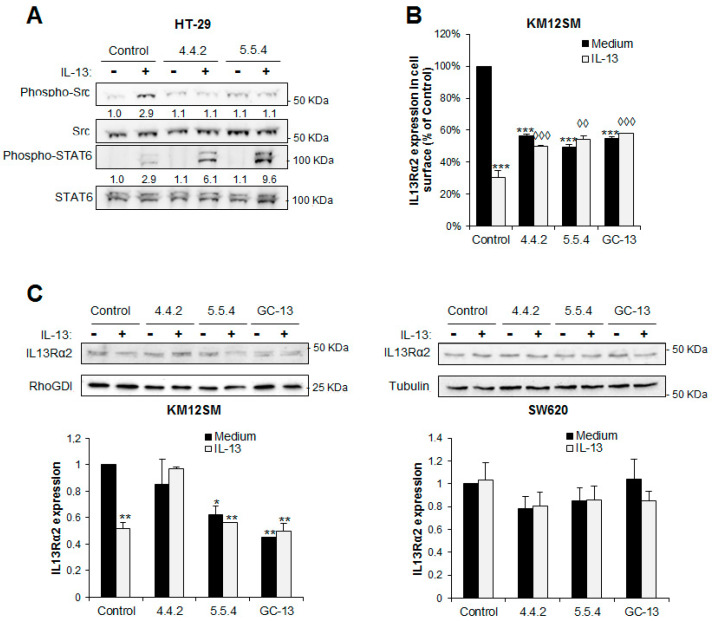
Effect of IL13Rα2 D1 mAbs on IL-13 signaling pathways, receptor internalization and degradation. (**A**) HT29 cells expressing IL13Rα1 and IL13Rα2 receptors were treated or not with IL-13 in presence of the IL13Rα2 D1-specific mAbs and then analyzed by western blot to detect phospho-SRC and phospho-STAT6. Total SRC and STAT6 were used as loading controls. Bands were quantified and normalized with respect to untreated control condition. Both mAbs caused a clear inhibition of SRC activation but not IL13Rα1-mediated STAT6. (**B**) Inhibition of IL-13-promoted receptor internalization by D1-specific mAbs. All D1-specific antibodies inhibited IL-13-induced receptor internalization at a significant extent (^◊◊^
*p* < 0.01; ^◊◊◊^
*p* < 0.001) and the internalization of the receptor by the mAbs compared with a control antibody (^***^
*p* < 0.001). (**C**) Inhibition of receptor degradation after IL13-promoted internalization by D1-specific mAbs in KM12SM and SW620 cells. IL13Rα2 degradation was detected by western blot and quantified by densitometric analysis (* *p* < 0.05; ** *p* < 0.01; *** *p* < 0.001).

**Figure 6 cancers-13-01731-f006:**
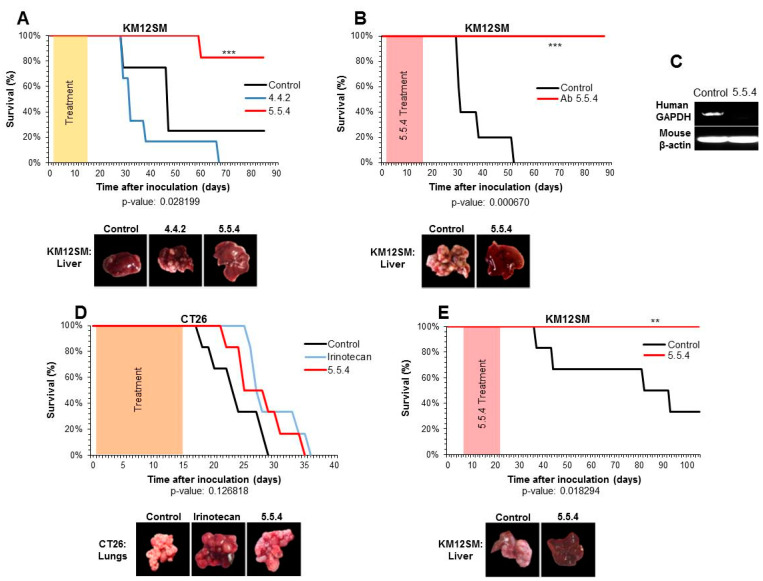
MAb 5.5.4 protects mice from liver metastasis induced by IL13Rα2-positive cancer cells. (**A**) Swiss nude mice were inoculated intrasplenically with metastatic KM12SM cells and 48 h later mice were treated with 4.4.2 and 5.5.4 mAbs (50 mg/Kg of weight, divided in 7 doses) administered intravenously. Kaplan–Meier survival results indicated a significantly enhanced survival after treatment with the 5.5.4 mAb (*** *p* < 0.001). Representative pictures of livers from the inoculated mice after necropsy are shown. (**B**) Swiss nude mice were treated as in A, but only mAb 5.5.4 was used for the treatment. Kaplan-Meier analysis indicated a complete survival of treated mice (*** *p* < 0.001). (**C**) To study cell homing, Swiss nude mice were inoculated intrasplenically with metastatic KM12SM cells in presence of 5.5.4 or a control antibody. After 96 h, mice were euthanized and mRNAs were isolated from livers and subjected to RT-PCR to amplify human GAPDH (and mouse β-actin as control). (**D**) Swiss nude mice were inoculated intra-tail vein with mouse CT-26 cells and 48 h later treated with 7 doses of mAb 5.5.4 or 4 doses of irinotecan. According to Kaplan-Meier analysis mice did not show a significant survival in any of the treated groups (*p* = 0.126). Finally, (**E**) Swiss nude mice were inoculated intrasplenically with metastatic KM12SM cells and 7 days later we started the treatment with 5.5.4 at the same dose. Treated mice showed a complete survival (** *p* < 0.01, *p* = 0.018).

## Data Availability

The data that support the findings of this study are available in the figures and the supplementary material of this article.

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
