# Peer review of "Inhibition of Liver Metastasis in Colorectal Cancer by Targeting IL-13/IL13Rα2 Binding Site with Specific Monoclonal Antibodies"

_cancers, 2021, doi:10.3390/cancers13071731_

Round 1

Reviewer 1 Report

  1. Fig 2A Sandwich ELISA: It was not clear from the associated text in Results section or the figure legend how the sandwich ELISA was done. Materials and Methods section provides the coverage, but it will be more reader friendly if such info could be also introduced, briefly, in Result. In addition, “an irrelevant antibody was used as a control” was stated in Methods but no such results seem shown in Fig 2A.

  1. An important biochemical characterization on binding kinetics directly between the isolated mAbs and IL13Ra2, e.g. kon and koff by BIAcore or Octet was missing.

  1. Also it is very important to test mAb’s binding selectivity on Ra2 over Ra1, as the later one broadly expresses among normal tissues.

  1. The signals reported in Fig 2B were relatively weak only at 0.1 OD450. Is it because only 0.1 ug / ml of IL-13 was used (as Methods state) to avoid high [mAb] needed to replace IL-13 binding on Ra2, as IL13-Ra2 gives fmol affinity? It really brings to the point of need to measure Kd of isolated mAbs (see comment #2).

  1. Even the mAbs were raised with peptides derived from D1 loop of s-Ig domain of Ra2, it likely but does not necessarily mean the epitope is located at D1. Competitive ELISA of mAbs on immobilized Ra2 in the presence of increasing concentrations of D1 peptide can give additional clues. Overall, the current submission has no efficient evidence to claim that these mAbs are “D1-specific”. Ideally, alanine mutations on D1 of Ra2 can give better answers (but maybe beyond the scope of current study). As Ra1 and Ra2 are different at D1, at least the selectivity test should be considered (see comment #3).

Reviewer 2 Report

Jaen et al. present an interesting study investigating the development of IL13Ralpha2 D1 peptide-specific monoclonal antibodies to block IL13Ralpha2 activation in vitro and treat colorectal cancer liver metastasis using a murine model. There is a clear clinical need to improve treatment of colorectal liver metastasis.

  1. In Figure 2C, please clarify what is meant by “lysate”.
  2. Figure 3 legends lack sufficient detail to easily interpret the results. Please clarify what comparisons are represented by the asterisks and the open diamonds. What statistical test was used? Was p<0.001 represented by three open diamonds, and not two as described in the legend. What was the “control” in these experiments? What was meant by “input” in the MTT assay?
  3. Figure 4 may have benefitted from quantification and normalisation of the bands.
  4. In Figure 5A do the numbers represent normalised pSRC and pSTAT6? In Figure 5C, how many replicates? The authors claim that the observed differences in IL13Ralpha2 degradation between KM12SM and SW620 are probably due to different expression levels - How was IL13Ralpha2 expression level determined?
  5. In Figure 6, please clarify how many mice are in each group for each experiment (ie panel A, B, C and D)? Does 5.5.4 shrink established KM12SM tumours or inhibit growth ie what is the size of the tumours at day 7 before treatment commences, and is there evidence of residual tumour, H&E or otherwise, after treatment?
  6. The authors claim that mAb 5.5.4 caused “complete” inhibition of liver metastasis (line 382), however without microscopic examination of the liver, or some other evidence, is this would appear to be an over-statement.
  7. In the discussion, the authors claim that RKO and CT-26 express low levels of IL13Ralpha2 (line 398). Although they cite Castle et al. for CT-26, how was this determined for RKO?
  8. The claim that mAb 5.5.4 is anti-metastatic (ie inhibits metastasis) is not supported by the experimental models used.

Minor edits, including

Line 73: “D1sequence-specific mAbs” should be “D1 sequence-specific mAbs”

Line 113: PBS abbreviation not defined in the first instance.

Line 134: TMB abbreviation not defined in the first instance, then appears in Line 144.

Line 154: 2mm EDTA should be “2 mM”

Line 160: alternate use of concentration units, was mM now mmol/L

Line 199: Traswell should be transwell

Line 452: fullstop missing

Round 2

Reviewer 1 Report

I still believe a BIAcore measurement is needed. 

Reviewer 2 Report

All of my comments have been addressed to my satisfaction.